# A Clinical Approach of Allergic Rhinitis in Children

**DOI:** 10.3390/children10091571

**Published:** 2023-09-19

**Authors:** Ioannis Goniotakis, Evanthia Perikleous, Sotirios Fouzas, Paschalis Steiropoulos, Emmanouil Paraskakis

**Affiliations:** 1Pediatric Respiratory Unit, Pediatric Department, University of Crete, 70013 Heraklion, Greece; ioannis.goniotakis@gmail.com (I.G.); paraskakis@yahoo.com (E.P.); 2Pediatric Emergency Department, General Hospital of Nicosia, 2031 Nicosia, Cyprus; evanthia.perikleous@shso.org.cy; 3Pediatric Respiratory Unit, University Hospital of Patras, 26504 Patras, Greece; sfouzas@gmail.com; 4Department of Pneumonology, Medical School, Democritus University of Thrace, 68100 Alexandroupolis, Greece

**Keywords:** allergic rhinitis, asthma, children

## Abstract

Allergic rhinitis is an important disease with a global footprint and a growing prevalence, affecting children and adults. Although it is commonly under-diagnosed and under-treated, it causes important social and economic effects (diminished quality of life, poor academic performance, escalated medical visits, heightened medication usage, and effects in other chronic conditions, e.g., asthma). It is characterized by distinctive, easily identifiable symptoms (sneezing, nasal discharge, nasal congestion, nasal–eye–palatal itching) and indirect accompanying indicators (fatigue and decreased school performance). The classification of allergic rhinitis hinges upon its nature and chronic distribution (seasonal or perennial) and its intensity, which spans from mild to moderate and severe. The diagnostic process primarily relies upon recognizing key clinical indicators, evaluating historical records, and considering risk factors. It is supported by abnormal laboratory findings, like in vitro allergen-specific IgE tests (enzyme immunoassay—EIA, chemiluminense immunoassay—CLIA) or in vivo skin prick tests for specific allergens. In the differential diagnosis, other chronic diseases manifesting with chronic rhinitis should be excluded (e.g., rhinosinusitis, chronic non-allergic rhinitis, rhinitis triggered by medications). The treatment of allergic rhinitis in children is mainly chronic and is focused on allergen exposure prevention, drug therapy, and immunotherapy in severe cases. Locally administered intranasal corticosteroids are the cornerstone of therapy. They are safe, effective, and have a favorable safety profile even during long-term use. Choosing a suitable intranasal corticosteroid drug with low systemic bioavailability makes long-term treatment even safer. Combinations of intranasal corticosteroids and H1 antihistamines are available in several countries and are widely used in more severe cases and the presence of year-round symptoms. Adding newer-generation oral H1-antihistamines broadens the available therapeutic inventory without significant effects compared to using previous-generation, once widely available, H1-antihistamines. Treatment of allergic rhinitis is complex and multi-dimensional, requiring an effective approach by a specialized group of specialized pediatricians, and is severely affected by the concurrent presence or development of other diseases in the spectrum of allergic diseases (conjunctivitis, asthma).

## 1. Introduction

Allergic rhinitis is one of the most prevalent medical conditions globally and typically persists throughout an individual’s lifetime. The reported frequency of allergic rhinitis by those affected ranges from 2% to 25% in children and from 1% to >40% in adults [1]. However, the prevalence of confirmed allergic rhinitis in European adults is estimated to be between 17% and 28.5% [2].

Classic symptoms of allergic rhinitis include persistent sneezing, watery nasal discharge, itching, and nasal congestion. Ocular symptoms are also common. Allergic conjunctivitis is associated with itching, eye redness, and tearing. Other potential symptoms include upper lip itching, postnasal drip, and coughing [3]. Allergic rhinitis is often linked with asthma, present in 15% to 38% of patients with allergic rhinitis. Nasal symptoms occur in 6% to 85% of patients with asthma [4]. Additionally, allergic rhinitis is a risk factor for asthma onset, and uncontrolled moderate-to-severe allergic rhinitis significantly affects asthma control [5].

Allergic rhinitis may seem innocuous compared to other medical conditions as it is not associated with high morbidity and mortality. However, the burden of the disease and its related socio-economic costs are substantial [6]. Allergic rhinitis diminishes the quality of life for many patients, disrupting their sleep quality and cognitive function and inducing irritability and fatigue. Moreover, it is associated with reduced school and work performance, especially during peak pollen periods [7]. Allergic rhinitis is a frequent reason for visits to family pediatricians, primary care facilities, and hospitals (emergency departments and routine outpatient clinics). The appropriate treatment of allergic rhinitis improves symptoms, quality of life, and performance at work and school [8].

## 2. Diagnosis

The clinical suspicion of allergic rhinitis is initially raised based on the symptoms, which include one or more of the following: persistent sneezing, rhinorrhea, itching, watery nasal discharge, nasal congestion, and hypogeusia. These symptoms develop within minutes to 1–2 h after exposure to an allergen [9]. Late manifestations of allergic rhinitis include nasal congestion, hyposmia, nasal hyperreactivity, and postnasal drip. Allergic rhinitis can be perennial or seasonal, depending on the causative allergen. However, this classification is more complex and universally applicable. The group of specialized experts known as “ARIA” (Allergic Rhinitis and its Impact on Asthma) has classified allergic rhinitis as persistent or intermittent based on the duration of symptoms [10]. A comprehensive and thorough clinical history is imperative for diagnosing and identifying the phenotype of allergic rhinitis in children to administer appropriate treatment and determine the best management approach for the patient [11].

Key elements from the medical history include the timing/location of allergic rhinitis symptoms’ onset, factors that alleviate or exacerbate symptoms (e.g., exposure to chemicals or smoke at home or school), and triggers for symptoms (pollen, household dust, molds, plants/trees, mites, animals). Identifying comorbidities is essential and can be achieved through detecting and evaluating clinical manifestations. The presence of coughing, wheezing, and breathing difficulty signifies asthma. Eye redness, tearing, and swollen eyelids indicate allergic conjunctivitis. Daily fatigue, snoring, and sleep apnea may be indications of sleep disturbances.

Furthermore, poor school performance, speech and learning disorders, decreased concentration, and excessive TV watching at high intensity can indirectly suggest hearing impairments due to chronic rhinosinusitis, secretory otitis media, or adenoid hypertrophy. Other conditions that should be excluded are eczema and oral allergy syndrome. The medical history should encompass a thorough assessment of how allergic rhinitis impacts the child’s daily life, school performance, and sleep, as well as information from the family history regarding the presence of allergic or immunological disorders [12]. 

The examination should include anthropometric measurements and a detailed clinical assessment of critical systems (skin, ears, nose, mouth, throat, and chest). General assessment is even more important in children with a history of long-term corticosteroid use to highlight the presence of drug-induced adverse effects. Typical abnormalities include Dennie-Morgan lines, the allergic salute, allergic shiners, and mouth breathing. All these clinical points are collectively characterized as the ‘allergic facies.’ Nasal examination provides significant information and can be performed through anterior rhinoscopy using a common otoscope. Often, this can reveal secretions (clear and watery in cases of allergic rhinitis, dense and discolored in cases of non-allergic or infectious rhinitis), polyps, foreign bodies, nasal septal deviation, nasal septal perforation, nasal creases, and nasal mucosal swelling. Endoscopic examination of the nose using a specialized endoscope is not obligatory during the investigation. However, it can be helpful when the child’s symptoms persist throughout the year to rule out anatomical anomalies or non-allergic inflammatory conditions (e.g., chronic rhinosinusitis with/without nasal polyps). Endoscopy is also valuable for assessing the patency of the nasal cavities, the size of adenoids, and the presence of nasal lesions. From a functional perspective, it is crucial to evaluate the airflow from the nasal cavities (nasal ventilation) to detect unilateral or bilateral obstruction [11].

Determining a child’s allergic sensitization pattern using skin prick testing (SPT) or laboratory methods for the in vitro measurement of allergen-specific IgE (EIA, CLIA) is particularly valuable for completing the diagnostic assessment and determining further therapeutic options, especially if immunotherapy is a potential consideration [13]. Skin prick tests are the main in vivo testing method recommended by international guidelines, as they are considered the most sensitive and specific test for detecting atopy. In these tests, a small amount of a natural or synthetically prepared allergen or mixture is used on the skin. Positive (histamine) and negative (diluent) controls are used during the test to confirm the proper execution of the procedure, assessing skin reactivity and excluding false positive results. However, a positive test result does not independently diagnose allergic rhinitis but should be evaluated based on the patient’s clinical history to establish a causative correlation [14].

The measurement of specific IgE in the serum constitutes an in vitro laboratory test for investigating the sensitization pattern [15]. The absence of positive in vitro allergen-specific IgE tests for the most common environmental allergens does not rule out the presence of allergic rhinitis, as sensitization to other uninvestigated allergens cannot be excluded. Furthermore, in cases of local allergic rhinitis, test results can be negative or weakly positive [16]. 

Advances in the field of laboratory allergology have enabled the development of new tests for the quantitative determination of specific IgE, both for individual allergens (ImmunoCAP) and for multiple allergens (range of 100–300 allergens) simultaneously (technologies like ImmunoCAP ISAC, ALEX test, and others), which have replaced traditional laboratory techniques (RAST). These tests have now been collectively categorized as ‘molecular allergology’. They are characterized by high sensitivity and specificity, regardless of the patient’s age and current medication, offering the opportunity for laboratory assessment of allergic rhinitis even under antihistamine treatment [17]. Conversely, the reliability of in vivo tests (SPT) is catalytically affected by anti-allergic treatment. Specific IgE determination tests are still valuable for detecting and differentiating cross-reactive sensitization to select appropriate allergen extracts for immunotherapy applications. In the future, developing new biomarkers for diagnosing and substantiating allergic rhinitis (e.g., circulating miRNAs, biochemical metabolites, and others) is essential [18].

In the context of a personalized approach to pharmaceutical treatment and appropriate medium- to long-term monitoring, accurate differential diagnosis is crucial. A thorough investigation is necessary in cases where the sensitization profile is incompatible with clinical symptoms throughout the year or in patients with atypical clinical manifestations. For example, imaging is not always essential. However, when there is suspicion of sinusitis that cannot be clinically diagnosed or excluded, it is recommended that a sinus CT scan should be conducted.

In terms of differential diagnosis, it is essential to exclude other diseases with similar symptoms. Infectious rhinitis often occurs in the context of upper respiratory viral infections. Moreover, the spectrum of non-allergic/non-infectious rhinitis (non-allergic rhinitis, NAR) encompasses conditions that potentially manifest as rhinitis (exposure to irritants such as cigarette smoke, environmental pollutants, hormonal dysfunction, e.g., subclinical hypothyroidism in older children, exposure to medications, vasomotor and idiopathic rhinitis). Additionally, in the differential diagnosis, considerations include local allergic rhinitis (LAR), mixed rhinitis, anatomical abnormalities, hypertrophy of adenoids, and other conditions that mimic chronic rhinitis in pediatric patients (primary immunodeficiencies, cystic fibrosis, primary ciliary dyskinesia) [11].

The presence of unilateral symptoms, isolated nasal obstruction, mucopurulent rhinorrhea, posterior nasal discharge with viscous mucus with/without anterior rhinorrhea, pain, epistaxis, and isolated anosmia should direct the differential diagnosis to other clinical entities. For instance, chronic mucopurulent rhinitis may suggest the presence of rhinosinusitis due to adenoid hypertrophy, anatomical anomalies, primary immunodeficiency, primary ciliary dyskinesia, or cystic fibrosis [11].

Local allergic rhinitis (LAR) is a phenotype characterized by nasal type 2 allergic immune response with local production of specific IgE without evidence of atopy from skin prick tests or specific IgE serum tests. LAR likely represents a subtype of the clinical entity formerly known as “non-allergic rhinitis with eosinophilia syndrome” (NARES). There are strong indications for the underdiagnosis of this clinical entity. Patients exhibit rhinitis symptoms, nasal hyperreactivity, and positive nasal provocation tests for aeroallergens. These tests can be performed in children over five years, following the guidelines of the European Academy of Allergy and Clinical Immunology (EAACI) [16].

The measurement of local IgE in nasal lavage fluid distinguishes LAR from NAR in the pediatric population and predicts the response to classical allergic rhinitis treatment [19]. The basophil activation test (BAT) could potentially be used to confirm LAR with high specificity, but it is conducted in a few specialized laboratories primarily as a research tool. NAR is considered more common in adults, though the diagnostic criteria for pediatric age are unclear. It is characterized by rhinitis without systemic allergic or infectious signs. EAACI recommends nasal endoscopy for detecting chronic rhinosinusitis with or without nasal polyps. However, it does not recommend local allergen challenge tests, nasal secretion culture, or the cytological examination of nasal secretions that could differentiate inflammatory causes from the neurogenic etiology of symptoms. During a cytological examination of nasal secretions, eosinophilic inflammation can be detected. Without systemic allergy, it can be attributed to allergic rhinitis, NARES, or intolerance to drugs (e.g., aspirin), foods, or food preservatives [19,20].

## 3. Allergic Rhinitis and Asthma

Galen, a pioneer in medical history, ventured into uncharted territory through delving into the intricate realms of the upper airway and sinuses, recognizing their integration into the broader respiratory system. An audacious hypothesis emerged from his work, proposing that rhinitis and asthma sprouted from secretions cascading from the brain toward the nasal passages and lungs. Centuries later, the notion of “United Airway Disease” (UAD) has found a contemporary home in medical discourse.

The “United Airway Disease” presents two main phenotypes: allergic (atopic or exogenous) and non-allergic (non-atopic or endogenous). Most children and over 50% of adults exhibit the allergic phenotype, in which the disease is associated with sensitization to allergens (presence of specific IgE in the serum or positive skin prick tests to common inhaled allergens) [21].

The release of antimicrobial and anti-inflammatory proteins and other molecules (lysozyme, lactoferrin, antioxidants, secretory IgA) from the nasal mucosa protects the lower airways from infectious agents and allergens. However, under pathological conditions, patients with allergic rhinitis experience a partial or complete loss of the protective function of the nose due to nasal congestion, as the nasal airways are bypassed during mouth breathing. Consequently, the lower airways are directly exposed to pathogens and allergens, increasing the likelihood of asthma. Additionally, the inhalation of cold and warm air following the loss of air temperature regulation by the nasal cavities can lead to immediate bronchoconstriction, a dominant feature of asthma [22].

The two diseases, allergic rhinitis and asthma, share common pathophysiological characteristics such as inflammation, anatomy, airway remodeling, environmental factors, and genetic predisposition. Regarding inflammation, it is usually eosinophilic. The presence of eosinophils in induced sputum of patients with intermittent allergic rhinitis, even during non-sensitization periods, as well as the presence of pathological findings in bronchial mucosa biopsies from patients with severe allergic rhinitis without clinical manifestations of bronchial asthma, provide supportive evidence for the concept of “United Airway Disease”. Additionally, patients with intermittent allergic rhinitis show increased fractional exhaled nitric oxide (FeNO) baseline levels, which increase even further during sensitization periods. Furthermore, patients with persistent allergic rhinitis exhibit higher FeNO values than patients with intermittent allergic rhinitis [23].

Allergic rhinitis has significant implications for childhood asthma, as it (a) constitutes a risk factor for the development of bronchial asthma, (b) affects asthma control, and (c) impacts the prognosis and progression of asthma into adulthood [24]. The presence of allergic rhinitis is a burdening factor for the persistence of childhood asthma into adulthood (OR, 2.7; 95% CI, 1.3–5.6). Moreover, among all patients with asthma, 29.7% of cases of persistent atopic asthma and 18.1% of newly onset asthma in adulthood can be attributed to allergic rhinitis or eczema in childhood. Based on studies in UK children, allergic rhinitis and asthma led to a noticeable surge in hospital admissions, medical appointments, and overall healthcare expenditures. Notably, among children, the likelihood of requiring hospital readmission due to asthma was notably greater for those suffering from AR and asthma than those solely dealing with asthma. As gleaned from the research mentioned earlier, the financial burden of caring for individuals with asthma and AR significantly outweighs those with asthma alone. The management of AR could potentially enhance the regulation of asthma symptoms and mitigate exacerbations. The significance of the association between asthma and AR/non-AR and the adverse effects of rhinitis on asthma control is underscored by the globally high and escalating prevalence of asthma and rhinitis [25].

While achieving guideline-defined asthma control might be feasible within the controlled confines of clinical trials, the reality outside those boundaries often paints a different picture. Once they enter the real world, many individuals with asthma continue to suffer from symptoms and with suboptimal asthma management. Approximately 20–60% of individuals with rhinitis concurrently exhibit clinical signs of asthma, while an overwhelming >80% of those dealing with allergic asthma also battle the burdens of rhinitis symptoms. Evidence gathered through pathophysiological exploration studies parallels the anatomical and physiological aspects of the nasal passages and bronchial pathways. The common triggers capable of causing exacerbations in both asthma and rhinitis, coupled with the shared inflammatory responses they induce, further emphasize their intertwining. Studies have even uncovered that allergen challenges in the upper airways can cause a significant decrease in lung function and bronchial responsiveness, further fueling the notion of a unified airway.

Recent studies in pharmacological interventions underline new important points. Mometasone furoate nasal spray, a low-bioavailability nasal corticosteroid, emerges as a valuable tool, having exhibited the ability to enhance the quality of life and alleviate the respiratory symptoms of children suffering from persistent allergic rhinitis and asthma. Rhinitis, acting as a precursor, emerges as one of the most potent standalone risk factors for the onset and progression of asthma. Childhood rhinitis sets the stage for an escalated risk of asthma in adolescence, adulthood, or middle age. Yet, not all sufferers perceive their rhinitis symptoms as disruptive forces in their social, academic, or professional lives, thus often avoiding medical intervention. The crossroads of treatment strategy becomes a complex juncture. The well-described pathways of individualized rhinitis and asthma treatments diverge when co-morbid conditions enter the scene. However, explorations into commonly employed pharmacological treatments illuminate a promising landscape. Intranasal glucocorticosteroids, antihistamines, anti-leukotrienes, and immunotherapy emerge as weapons, potentially ameliorating both rhinitis and asthma symptoms for those children suffering from both diseases. Guiding patients toward improved compliance with treatment emerges as a promising strategy. The regular monitoring of treatment adherence remains pivotal until an optimal level of symptom control is attained. A key element lies in clear communication, equipping patients and parents of young children with an understanding of all available treatment options and strictly following a stable approach to management [26].

The fundamental principles of optimal AR therapy are applicable regardless of whether an individual has concomitant asthma. Detecting the presence of AR should lead to identifying and addressing triggers and irritants that can impede the management of both asthma and AR. Asthma and AR frequently co-occur within the same individual, necessitating a comprehensive evaluation and dual management approach. Every child, adolescent, and adult with asthma warrants an assessment for AR through targeted inquiries concerning the nature, frequency, and severity of nasal symptoms. Particularly for individuals already diagnosed with AR, especially if it is persistent, an evaluation of asthma is equally important. This assessment might involve questions related to wheezing, coughing, and breathlessness during physical activity but could also require pre- and post-bronchodilator spirometry measurements for individuals aged 7–8 years and older. It’s important to note that addressing AR is often an integral step toward effective asthma control, and successful management might remain elusive until the recognition and treatment of AR are adequately addressed. Despite genetic predisposing factors, the impact of allergic rhinitis on asthma can be controlled with timely, aggressive treatment of rhinitis. It is critical to evaluate asthma in every child diagnosed with allergic rhinitis during follow-up and treatment [27,28,29].

## 4. Treatment

The treatment of allergic rhinitis is based on four main axes: (a) allergen avoidance; (b) systematic nasal lavage (saline nasal irrigation); (c) the administration of targeted, effective, and clinically responsive therapy; and (d) the implementation of immunotherapy [30].

For patients with seasonal allergic rhinitis, recommended first-line treatment options are (1) intranasal corticosteroid administration with or without concomitant oral H1 antihistamines or (2) intranasal corticosteroid administration combined with oral H1 antihistamines. The combination of intranasal corticosteroids and intranasal H1 antihistamines results in a faster response compared to the isolated use of intranasal corticosteroids. Response assessment is recommended after 2–4 weeks to determine the further therapeutic plan. If there is a response, treatment should be continued for at least one month. In case of exacerbation, escalation of the administered treatment is necessary. Second-line treatments can include oral H1 antihistamines or anti-leukotrienes [31].

For patients with perennial allergic rhinitis, recommended first-line treatment options, based on preferences and availability, are (1) monotherapy with intranasal corticosteroids or (2) intranasal corticosteroid administration combined with oral H1 antihistamines. In cases of perennial allergic rhinitis, oral H1 antihistamines administered orally are recommended as second-line treatment, while oral leukotriene receptor antagonists are considered third-line treatment [32,33].

Intranasal corticosteroids have a good safety profile regarding long-term adverse effects. However, choosing the appropriate type of corticosteroid is vital due to significant differences in absorption from nasal mucosa and systemic bioavailability. Fluticasone formulations (furoate, propionate) and mometasone are preferred as they exhibit minimal systemic absorption, reducing the risk of systemic corticosteroid complications [32,33]. Concerning orally administered H1 antihistamines, older-generation antihistamines (e.g., hydroxyzine) are contraindicated due to the availability of newer, safer H1 antihistamine formulations. First-generation antihistamines effectively cross the blood–brain barrier and affect brain function, inducing sedation, cognitive impairment, and psychomotor development. Their widespread over-the-counter distribution during the previous decades has highlighted the possibility of serious, even fatal, side effects during overdose and “off-label” usage as sedatives/anxiolytics (e.g., respiratory depression, seizures, coma, death) [34]. Newer-generation formulations can be safely used for an extended period, including >1 year of continuous use in preschool-aged children [34].

Saline nasal irrigation (SNI) using isotonic solution is used as an adjunctive non-pharmacologic treatment for allergic rhinitis. It has proven efficacy in acute and chronic rhinosinusitis, as well as in cases after sinus surgery. Systematic reviews of the literature suggest that it is a useful additional treatment for AR with low cost, easy application, and infrequent side effects [35].

The use of allergen immunotherapy (AIT) is an alternative solution in cases of severe non-responsive rhinitis to first- to third-line treatments or for reducing the risk of progression to asthma in children with allergic rhinitis and atopic dermatitis. This proposal is attractive based on the concept of the atopic march, where a child with eczema and food allergies develops asthma and allergic rhinitis in later childhood. However, systematic literature reviews provide limited evidence supporting the use of AIT as a preventive intervention [36,37,38]. There seems to be short-term benefit in preventing asthma in individuals with allergic rhinitis and grass and ragweed sensitization, especially when AIT is initiated in childhood using subcutaneous immunotherapy (SCIT) and sublingual immunotherapy (SLIT) [36]. Data are insufficient to support or discourage the use of AIT against other allergens (mites and others.). Evidence of the effectiveness of SCIT in children is scarce compared to adults. Despite this, AIT is the only targeted allergen tolerance-inducing treatment currently available. The decision to initiate allergen immunotherapy depends on several factors, such as the severity of the disease, the expected adherence to therapy, the child and family background, the expected results, a thorough understanding of possible side effects, the high cost and duration of treatment, and results [37]. The drawbacks of this intervention should be thoroughly discussed with the family before treatment initiation. SCIT administration is particularly challenging for children because of multiple injections. SLIT is a more friendly approach for children. Still, it is less effective in managing allergic rhinitis caused by common aeroallergens. However, it is still being determined whether this benefit persists for several years after discontinuation of AIT or if AIT is indeed a cost-effective intervention [38].

## 5. Conclusions

In conclusion, allergic rhinitis is a significant medical condition in childhood, adolescence, and adulthood. The hallmark symptoms of persistent sneezing, watery nasal discharge, itching, and nasal congestion are accompanied by ocular manifestations, including itching, eye redness, and tearing, alongside other potential symptoms like upper lip itching, postnasal drip, and coughing. A web of interconnectedness binds allergic rhinitis and asthma, with a prevalence of nasal symptoms in asthma patients and vice versa. The role of allergic rhinitis as a risk factor for asthma onset and its potential to compromise asthma control are underscored. While allergic rhinitis might appear less important regarding mortality and morbidity, its burden extends beyond its clinical spectrum. It impacts the quality of life, sleep patterns, cognitive function, and overall performance, amplifying the societal and economic costs. Its significance as a reason for healthcare visits, especially among pediatric patients, underscores the need for proper management and treatment.

The diagnostic approach is based on clinical manifestations and medical history, relying on targeted questions and examination to unveil the presence of allergic rhinitis, its triggers, and potential comorbidities. The differentiation between infectious and non-allergic causes and the recognition of local allergic rhinitis necessitates thorough evaluation. The interplay between allergic rhinitis and asthma unveils the “United Airway Disease” concept, emphasizing the common features of inflammation, anatomy, airway remodeling, and genetic factors. This relationship pinpoints the impact of allergic rhinitis on asthma development, control, and outcomes.

Treatment strategies include allergen avoidance, saline nasal irrigation, pharmacotherapy, and potential immunotherapy. Tailored interventions based on symptom patterns and allergen sensitivities are pivotal. Intranasal corticosteroids emerge as a mainstay of treatment, complemented by antihistamines, anti-leukotrienes, and, in certain cases, immunotherapy. The management pathway underscores the significance of addressing allergic rhinitis to enhance asthma control.

Allergic rhinitis’s significance extends far beyond its seemingly innocuous nature. The symptoms and their association with asthma underscore its clinical importance, guiding medical professionals toward a comprehensive approach to diagnosis and management.

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
