# Peer review of "A Clinical Approach of Allergic Rhinitis in Children"

_children, 2023, doi:10.3390/children10091571_

Round 1

Reviewer 1 Report

This is a general review of pediatric allergic rhinitis, and the content is well organized. I would like you to add some clinically useful content on several points.

1. I think the recommendation level for antihistamines that have sedative effects should be clarified. I would also like a detailed explanation regarding the issue of side effects. When using sedative drugs in children, there should be more problems than adults, such as seizures and impaired performance.

2. Are there any tips for successfully continuing allergen immunotherapy for children? I think that there are cases where clinicians are hesitant to introduce it because it is more difficult to continue treatment than in adults. Additionally, side effects may be a problem when performing allergen immunotherapy. Please describe specific measures to be taken to prevent side effects in children.

Author Response

We would like to thank you very much for the acceptance of our manuscript with minor revisions and for your insightful comments. We have uploaded the revised manuscript based on your comments.

Comments to reviewer 1

Thank you very much for your useful review and acceptance of our article. We have clarified our recommendation against the use of 1st generation antihistamines mentioning the possible side effects (line 322-328). We have also added an additional reference to our statements (line 461-462). AIT is a special matter of discussion and matter of on-going research in pediatric patients. Challenges in introduction and continuation of this intervention in children were added in the text (line 340-351). We have also added an additional reference (line 469-470).

Reviewer 2 Report

The manuscript is presented in an understandable manner and written in standard English which, however, should be proofread.

The text is well structured and covers the topic of major interest. The authors have presented a comprehensive and systematic review of allergic rhinitis in children, a diagnosis that has often been missed or neglected, in adults as well as in children..

 However, I have a few suggestions and comments:

Lines 22, 108 – “RAST “, which is short for radioallergosorbent test, is not a synonym for  determining the concentration of specific IgE antibodies. It reffers to a radioimmunoassay technique which was the first in vitro qualitative test for detecting and measuring circulating IgE antibodies. However, RAST is nowadays only rarely used because of the newer and more accurate methods.( eg. Enzyme Immune Assay or Chemiluminescence).

Line 107 – If the authors want to describe a skin prick test, then it is necessary to state a few more facts:  the standard set of inhalant allergens that is used in a case of suspected allergic rhinitis,  drugs that affect the skin reactivity and interpretation of SPT, measurement and interpretation of SPT, etc. It would be better to  just cite the literature that describes the method.

Line 116 – Positive and negative controls are mandatory to assess the skin reactivity and exclude false positive results of skin prick test.

Lines 120-128 – It is not necessary to describe the method in details, citation of the literature that describes the method (or metrhods) of determining the concentration of specific IgE antibodies would be better.

Line 130 – Local allergic rhinitis is explained later and it is not necessary to mention it here.

Line 150 - “conducting”  is not necessary

Line 152 – Instead of  "During the process of differential diagnosis, is” - I propose – “It is essential to exclude other conditions and diseases with similar symptoms."

For better understanding and clarity, I suggest listing the conditions which have to be considered in differential diagnosis, in table.  

Line 291 – “b) systemic nasal lavage” -  What did the authors mean by this? It is neither explained in the text nor is the  treatment recommended in guidelines. 

Lines 293-296  - I suggest to rephrase the sentence: “ For patients with seasonal allergic rhinitis, recommended first-line treatment…”.

Line 303 – I suggest to rephrase the sentence: “ For patients with perennial allergic rhinitis, recommended first-line treatment..."

Line 327 – “particularly” is redundant, since only SLIT and SCIT are applied in the treatment of AR.

Line 331 – The GAP Study has shown  that benefit of AIT on clinical symptoms of AR and preventive effect on development of asthma persist several years after discontinuation of AIT. (Valovirta E at al. JACI 2017).

Line 354  - “nasal lavage”??

Minor editing of English language is required.

Author Response

We would like to thank you very much for the acceptance of our manuscript with minor revisions and for your insightful comments. We have uploaded the revised manuscript based on your comments.

Comments to reviewer 2

Thank you very much for your useful review and acceptance of our article. Minor language editing was performed, as requested (highlighted text).

Line 22, 108. “RAST” was removed and newer methods (EIA, CLIA) were added.

Line 107. Proper citation was added. Text removed.

Line 116. Text added according to suggestions.

Line 120-128. Citation was added.

Line 150. Text modified.

Line 152. Thank you for your suggestions. We have modified the text accordingly. 

Line 291. Text modified – data added / reference added (line 321-329, line 474-475).
Line 293-296. Thank you for your suggestion. Text modified.

Line 303. Rephrased.

Line 327. “Particularly” was removed. New location line 338-339.

Line 331. Thank you for the well-taken point. We are aware of the study. The problem is that these results are not confirmed with other aeroallergens and this is the reason to why immunotherapy is not currently suggested for asthma.

Line 354 – rephrased (saline nasal irrigation)